# Potential Onco-Suppressive Role of miR122 and miR144 in Uveal Melanoma through ADAM10 and C-Met Inhibition

**DOI:** 10.3390/cancers12061468

**Published:** 2020-06-04

**Authors:** Adriana Amaro, Michela Croce, Silvano Ferrini, Gaia Barisione, Marina Gualco, Patrizia Perri, Ulrich Pfeffer, Martine J. Jager, Sarah E. Coupland, Carlo Mosci, Gilberto Filaci, Marina Fabbi, Paola Queirolo, Rosaria Gangemi

**Affiliations:** 1IRCCS Ospedale Policlinico San Martino, 16132 Genova, Italy; adriana.amaro@hsanmartino.it (A.A.); michela.croce@hsanmartino.it (M.C.); silva.ferrini@gmail.com (S.F.); gaia.barisione@gmail.com (G.B.); ulrich.pfeffer@hsanmartino.it (U.P.); gfilaci@unige.it (G.F.); marina.fabbi@hsanmartino.it (M.F.); 2Azienda Sociosanitaria Ligure 3 (ASL3), Sistema Sanitario Regionale Liguria, 16149 Genova, Italy; marina.gualco@asl3.liguria.it; 3Istituto di Ricovero e Cura a Carattere Scientifico (IRCCS) Istituto G. Gaslini|Ospedale Pediatrico Gaslini, 16147 Genova, Italy; patriziaperri@gaslini.org; 4Leiden University Medical Center, 2333ZA Leiden, The Netherlands; m.j.jager@lumc.nl; 5Department of Molecular and Clinical Cancer Medicine, University of Liverpool, Liverpool L7 8TX, UK; s.e.coupland@liverpool.ac.uk; 6Liverpool University Hospitals NHS Foundation Trust, Liverpool L7 8XP, UK; 7Galliera Hospital, 16128 Genova, Italy; carlomosci@icloud.com; 8Centre of Excellence for Biomedical Research and Department of Internal Medicine, University of Genoa, Via De Toni 14, 16132 Genova, Italy; 9IEO Institute of Oncology Istituto di Ricovero e Cura a Carattere Scientifico (IRCCS), 20141 Milan, Italy; paola.queirolo@ieo.it

**Keywords:** uveal melanoma, miRNA, ADAM10, c-Met, onco-suppressor

## Abstract

Uveal melanoma (UM) is a rare tumor of the eye that leads to deadly metastases in about half of the patients. ADAM10 correlates with c-Met expression in UM and high levels of both molecules are related to the development of metastases. MiR122 and miR144 modulate *ADAM10* and *c-Met* expression in different settings. We hypothesized a potential onco-suppressive role for miR122 and miR144 through modulation of ADAM10 and c-Met in UM. We analyzed the UM Cancer Genome Atlas data portal (TCGA) dataset, two other cohorts of primary tumors and five human UM cell lines for miR122 and miR144 expression by miR microarray, RT-qPCR, Western blotting, miR transfection and luciferase reporter assay. Our results indicate that miR122 and miR144 are expressed at low levels in the UM cell lines and in the TCGA UM dataset and were down-modulated in a cohort of seven UM samples, compared to normal choroid. Both miR122 and miR144 directly targeted *ADAM10* and *c-Met*. Overexpression of miR122 and miR144 led to reduced expression of ADAM10 and c-Met in the UM cell lines and impaired cell proliferation, migration, cell cycle and shedding of c-Met ecto-domain. Our results show that miR122 and miR144 display an onco-suppressive role in UM through ADAM10 and c-Met modulation.

## 1. Introduction

Uveal melanoma (UM) is a rare tumor of the eye that gives rise to metastases, especially in the liver, and is deadly in about one half of cases. Metastases are insensitive to chemotherapy, thereby leading to a high mortality rate [1,2,3]. Mutually-exclusive mutations in the *GNAQ* or *GNA11* genes, which encode signaling proteins associated with G-protein-coupled receptors, are specific drivers of UM [4,5]. Genetic alterations in primary UM such as monosomy of chromosome 3 and amplification of chromosome 8q are related to a poor prognosis [6,7,8]. In addition, inactivating mutations in the *BAP1* gene, which is located on chromosome 3, confer a more aggressive behavior to UM [9,10,11]. In contrast, mutations in *TP53*, *BRAF*, *NRAS*, *TERT* and the *CDKN2* genes, which are common in cutaneous melanoma [12,13], are rare in UM [1]. Besides mutated proteins, high expression of a disintegrin metalloproteinase 10 (ADAM10) and the tyrosine-kinase receptor c-Met [14,15,16] has been associated with UM progression. ADAM10 is a membrane-associated sheddase and has been involved in the proteolysis of several protein substrates and in tumor spread in several types of cancer [17]. c-Met, the receptor for the hepatocyte growth factor, mediates tumor cell invasiveness and has also been correlated with oncologic progression [18]. Similarly, we found that expression of *ADAM10* and *c-Met* mRNA was correlated with the development of metastases in a cohort of 108 primary UM [14]. In addition, active ADAM10 is present in UM cell lines and the ADAM10 substrate c-Met is shed as ecto-domain into the tissue culture medium. Indeed, we identified soluble c-Met as a potential biomarker in the sera of metastatic UM patients [19]. The finding that silencing of *ADAM10* reduces UM cell invasion further supports an important role for ADAM10 in UM progression and suggests that ADAM10 may be a potential target for therapeutic intervention [14].

MicroRNAs (miRs) are a class of small, non-coding, single-stranded RNAs that exert a post-transcriptional control of gene expression. MiRs preferentially target the 3’-untranslated region (UTR) of specific sets of mRNA [20] and modulate various biological processes such as proliferation, cell cycling, differentiation, apoptosis and epithelial-mesenchymal transition. Several miRs have been associated with ADAM10 and c-Met: previous reports showed that miR122 down-regulates ADAM10 expression in breast cancer [21] and hepatocellular carcinoma [22], while miR144 may play a role in down-regulating ADAM10 expression in Alzheimer disease [23]. MiR122 is highly expressed in the normal liver, whereas it is down-regulated in hepatocellular carcinoma [24,25]. In addition, miR122 was reported as a potential onco-suppressor molecule in non-small cell lung cancer [26], gallbladder carcinoma [27], bladder cancer [28], breast cancer [21] and gastric cancer [29,30]. Moreover, it was recently reported that miR122 plays a role in hepatocellular carcinoma by directly inhibiting c-Met expression [31]. Additionally, the expression of miR144 is significantly down-regulated in different cancers including gastric [32], breast [33], hepatocellular carcinoma [34] and cervical cancer [35]. MiR144 was found to be decreased in UM and restoration of its expression reduced in vitro proliferation and invasion of UM cells by directly targeting the 3’ UTR of *c-Met* [36]. Both miR122 and miR144 are involved in the post-transcriptional regulation of ADAM10 and c-Met and their down-regulation may thus contribute to the high expression of ADAM10 and c-Met seen in UM. Other studies have also examined miR expression in UM [37,38,39], but hardly ever addressed the aforementioned two and functional studies were limited to a few selected miRs identified in cell lines or in limited sample collections of UM [37]. Here, we investigate the role of miR122 and miR144 in the modulation of ADAM10 and c-Met expression and their influence on proliferation and invasiveness of UM cell lines. We show, for the first time, that miR144 and miR122 inhibit both c-Met and ADAM10 expression in UM cells.

## 2. Results

### 2.1. Analysis of miR122 and miR144 Expression in Primary UM Tumors and UM Cell Lines

To address the potential role of miR122 and miR144 in UM, we first analyzed the miR Cancer Genome Atlas data portal (TCGA) dataset that includes 78 primary UM bearing *GNAQ* or *GNA11* mutations. MiR144 and miR122 belonged to the lowest (0–30%) percentile of the global miR expression (Figure 1). Other miRs which could potentially target *ADAM10* or *c-MET* (based on the Miranda target prediction program), such as miR34a-5p [40], miR34c-5p [41] and miR140-5p [42,43], showed levels of expression above the 70th percentile (Figure 1). MiR122 and miR144 showed a low expression not only in the UM TCGA dataset but also in an independent cohort of 19 primary UM that were analyzed for miR expression by microarray analysis (Figure 2A,B). In addition, miR221 and miR222, which are known to be expressed in cutaneous melanoma [44], showed a high expression (Figure 1 and Figure 2A,B) and were then arbitrarily chosen as controls for highly-expressed miRs. No significant differences in miR122 and miR144 expression were found between metastatic and non-metastatic patients in these two datasets (miR122 *p* = 0.66 and miR144 *p* = 0.77).

To further corroborate these data, we analyzed the expression of miR122 and miR144 by RT-qPCR in seven additional primary UM and in normal choroid. U6 small nuclear RNA was used as an internal control, while miR221 and miR222 were analyzed as controls. The expression of both miR144 and miR122 was lower in the primary UM than in the normal choroid (Figure 2C).

Finally, we analyzed miR122 and miR144 by RT-qPCR in a panel of five UM cell lines (92.1 [45], MEL270, OMM2.5 [46], UPMM2 and UPMM3 [47]). OMM2.5 is derived from a liver metastasis from the same patient as MEL270 whereas UPMM3 and UPMM2 are derived from primary UM monosomic for chromosome 3. MiR122 and miR144 showed a very low expression relative to miR221 and miR222 in all the cell lines (Figure 2D).

Taken together, these data indicate a low expression of miR122 and miR144 in both primary UM and UM cell lines.

We next investigated whether there was a difference in *c-Met* and *ADAM10* expression in patients with higher miR122 and miR144 compared to patients with lower miR122 and miR144. Analysis of the TCGA dataset revealed that low expression (bottom quartile) of both miRs was associated with higher expression of both *ADAM10* and *c-Met*. This difference was statistically significant for miR144 (Appendix A).

### 2.2. Effect of Synthetic miR122 and miR144 Transfection on ADAM10 and c-Met Expression

To investigate the effects of miR122 and miR144 over-expression in UM cells we transfected miR-mimics into the 92.1 and UPMM3 cell lines, which represent different types of primary UM. Indeed, the 92.1 cell line is disomic for chromosome 3 and *BAP1* wild-type, whereas UPMM3 is monosomic for chromosome-3 and *BAP1* mutated [45,47]. RT-qPCR showed an increase of the specific miR122 or miR144 in cells transfected with the corresponding miR mimic, indicating efficient intracellular delivery of miRs (Figure 3A,B). MiR144 expression increased much more upon transfection than miR122, particularly in cell line UPMM3. RT-qPCR analysis of *ADAM10* and *c-Met* mRNA expression in miR-transfected cells revealed that miR122 inhibits not only mRNA expression of *ADAM10*, as found in other cellular models, but also of *c-Met* (Figure 3C,D) in both UM cell lines. Similarly, miR144 reduced both *c-Met* and *ADAM10* mRNA levels (Figure 3C,D). Indeed, miR122 and miR144 show complementary sequences with both *ADAM10* and *c-Met* 3’-UTR mRNA (Appendix A).

We next analyzed the effect of mimic miR transfection on protein expression. Western blot analysis showed that miR122 inhibited ADAM10 and c-Met protein expression in both 92.1 cells, and UPMM3 cells. Furthermore, transfection of miR144 reduced ADAM10 and c-Met protein levels, suggesting that both miRs can regulate ADAM10 and c-Met protein expression (Figure 4, Appendix A). Similar results were obtained using OMM2.5 cells, a cell line derived from a UM liver metastasis (Appendix A).

### 2.3. Effects of miR122 and miR144 on UM Cell Proliferation and Migration In Vitro

We and others have shown that ADAM10 and c-Met are related to UM invasiveness and progression [14,15]. Therefore, we explored the possible effects of miR122 or miR144 on UM cell proliferation and migration.

As shown in Figure 5A, transfection of miR144 reduced proliferation of 92.1 cells measured by a standard 3-(4,5-dimethylthiazol-2-yl)-2, 5-diphenyl tetrazolium bromide (MTT) assay, at every time point, whereas transfection of miR122 decreased 92.1 cell proliferation at 96 h. Overexpression of miR122 or miR144 mimic reduced UPMM3 cell proliferation at 96 h (Appendix A).

Consistent with the pro-invasive role of ADAM10 and c-Met, both miR122 and miR144 inhibited the migration of 92.1 cells, in response to Hepatocyte Growth Factor (HGF) as chemo-attractant, in an in vitro trans-well system (Figure 5B). In the UPMM3 cell model, miR144 significantly inhibited cell migration, while miR122 was less effective (Figure 5C).

Since our previous studies showed that ADAM10 processes surface c-Met and mediates c-Met ecto-domain shedding [19], we studied the possible effects of miR122 and miR144 mimic transfection on the release of soluble c-Met in the culture supernatant. As shown in Figure 5D, transfection of both miR mimics inhibited release of soluble c-Met from the 92.1 cells, while only miR144 significantly reduced soluble c-Met release in the UPMM3 cell line.

### 2.4. Effects of miR122 and miR144 on UM Apoptosis and Cell Cycle

We demonstrated that miR122 and miR144 overexpression reduced UM cell proliferation. However, MTT assay provides collective effects of cell proliferation and cell death. We therefore tested the possible role of these miRs in apoptosis and cell cycle modulation. Transfection of miR144 significantly increased the percentage of apoptosis in 92.1 cells, but not in UPMM3 cells. MiR122 did not affect apoptotic cell death in both cell lines (Figure 6).

Furthermore, Flow-cytometric analysis showed that miR144 mimic transfection significantly increased the proportion of cells in the G0/G1 phase and reduced the proportion of cells in the G2/M phase in comparison with irrelevant miR (Figure 7).

### 2.5. miR122 and miR144 Directly Target 3’UTR Region of Both ADAM10 and c-Met

To assess whether miR122 and miR144 directly down-regulate *ADAM10* and *c-Met* expression, we transfected 92.1 cells with the 3’UTR region of *ADAM10* and *c-Met* cloned in a plasmid vector, with luciferase as reporter gene. The luciferase reporter gene assay demonstrated that indeed both miR122 and miR144, co-transfected with pGL3-c-Met-3’UTR or pGL3-ADAM10-3’UTR, reduced luciferase activity, suggesting that miR122 and miR144 directly target the *ADAM10* and *c-Met* 3’UTR. Irrelevant control miR did not affect either pGL3-c-Met-3’UTR or pGL3-ADAM10-3’UTR luciferase activity (Figure 8).

## 3. Discussion

Metastatic UM still shows a high mortality rate, despite improvements in an efficient treatment of the primary tumor, in most cases. Hence, the identification and better understanding of the mechanisms involved in UM dissemination and colonization, together with the identification of new potential therapeutic targets and tools, is urgent. We have previously shown that both ADAM10 and c-Met are significantly more expressed in primary UM of patients undergoing liver metastases during follow-up, compared to those with good prognosis [14]. Interestingly, the expression of these two pro-metastatic genes is highly correlated [14], suggesting the possibility of a common mechanism regulating their gene expression. MiRs have been reported to contribute to cancer progression through the modulation of oncogenes or tumor suppressors. Alteration of several miRs has been described as potentially pathogenic in UM. Recently, miR181b was shown to be overexpressed in UM and to promote cell cycle progression in UM cells [48]. On the other hand, other miRs, such as miR9 [49], miR182 [50], miR34b and miR34c [41], may act as tumor suppressors in UM. In addition, miR144 [36] has been proposed as a tumor suppressor in UM by targeting the c-Met oncogene.

The present study shows that miR122 and miR144 are expressed at low levels in three different cohorts of primary UM and also in a panel of UM cells. Low levels of miR122 and miR144 in primary tumors may contribute to the up-regulation of both pro-metastatic genes *ADAM10* and *c-Met* and may function as tumor suppressors in this disease. Indeed, we found that low expression of miR122 and miR144 was associated with higher expression of both *ADAM10* and *c-Met* in the UM TCGA dataset. Furthermore, extremely low levels of expression of miR122 and miR144 were seen in UM cell lines, which are characterized by elevated levels of ADAM10 and c-Met [14,19]. Transfection of miR122 and miR144 mimics in 92.1 and UPMM3 cells leads to a reduction of *ADAM10* and *c-Met* mRNA and protein expression. Consistently with these data, we observed a reduction of proliferation, migration and shedding of c-Met in UM miR-transfected cells. In addition, an increase in G0/G1 phase of the cell cycle was present in both cell lines, while only 92.1 cells showed a concomitant increase of apoptosis. MiR122 has previously been reported to inhibit growth of hepatocellular carcinoma cells in vitro and in vivo [22] through the inhibition of different genes, including *ADAM10*. MiR122 also targets the 3’UTR of c-Met and its overexpression reduces proliferation and invasion of hepatocellular carcinoma cell lines [31]. The present work shows that both miR122 and miR144 overexpression inhibit ADAM10 and c-Met expression and have an anti-migratory and an anti-proliferative effect on the 92.1 and UPMM3 UM cells, although in the latter the effects are detected only at later time points. These effects seem to be related to a direct targeting of miR122 and miR144 on the 3’UTR of both *c-Met* and *ADAM10*. The effects of miR122 on ADAM10 and c-Met expression and on the proliferative and migratory activity of UPMM3 cells was less evident. The lower efficacy of miR122 in the UPMM3 model may relate to the low expression of this miR upon transfection, a finding that was consistently reproduced in three different transfection experiments. However, this low expression does not appear to be related to a generally low transfection efficiency of UPMM3 cells, as miR144 is well expressed upon transfection in these cells. In addition, transfection of fluorescent double strand RNA oligonucleotides produced comparable results in 92.1, OMM2.5 and UPMM3 cell lines (Appendix A). Recent findings indicate that endogenous transcripts control specific miRNA levels in mammalian cells by target-directed miRNA degradation [51]. One may speculate that similar mechanisms of enhanced degradation may inhibit endogenous or exogenous miR122 expression and activity in UM cells, particularly in the UPMM3 model. Differences in the functional effects of the two miRs may be also related to a different efficiency in reducing the targeted genes required to produce a functional effect, or to other possible additional targets of miR122 and miR144 not considered in this study. Chromosomal aberrations and *BAP1* mutation in UPMM3 cells may influence the onco-suppressive activity of miR122 and miR144. However, the analysis of miR expression in UM TCGA [6] found a cluster of miRs associated with monosomy 3 and *BAP1* mutation, but miR122/144 are not present in this cluster, and thus may not be regulated by them.

Finally, the more efficient ability of miR144 to block cell proliferation, stimulate apoptosis and arrest cell cycle implies an important role for c-Met and ADAM10 in these biological processes in UM. If miR144 [36] has already been proposed as a tumor suppressor by targeting the *c-Met* oncogene, its role on targeting *ADAM10* has not been demonstrated until now in UM. The effects of miR144 targeting both *c-Met* and *ADAM10* are evident in both disomic and monosomic cells, with a prevalent effect on disomic cell line.

Our data provide further information on the influence of ADAM10 and c-Met in the biology of UM and clarify some of the molecular mechanisms related to the modulation of miRs targeting the pro-neoplastic genes *ADAM10* and *c-Met*. The possibility to counteract pro-metastatic activity of onco-miRs has been investigated by several studies using different types of delivery tools [52,53]. Local delivery of miR122 and miR144 may be considered to restore tumor suppressive activity in UM and deserves to be further investigated in xenograft animal models of metastatic disease.

## 4. Methods

### 4.1. Cell Lines

Human UM cell lines 92.1 [45], MEL270, OMM2.5 [46], UPMM3 and UPMM2 [47] were cultured in Roswell Park Memorial Institute 1640 medium (Gibco/Life Technologies, Cheshire, UK), supplemented with 10% fetal bovine serum.

### 4.2. Patients and Tissue Samples

Normal choroid was obtained from a patient who underwent enucleation due to a basal-cell epithelioma of the eyelid, which had invaded the orbital cavity, and left the choroid intact. Informed consent had been obtained, in accordance with the Declaration of Helsinki. Tissue samples were obtained from 7 UM patients after enucleation surgery (Comitato Etico Regionale 12-2011, Emendamento sostanziale n.1 al Protocollo versione 2, 01-10/05/16). Patients and tumor characteristics have previously been described [14]. Five of the seven patients became metastatic during follow-up.

Another cohort of 19 primary UM derived from 5 metastatic and 14 non-metastatic UM patients was also used for miRs microarray analysis. For these UM patients, all human samples were provided by the Health Regulatory Authority (HRA) approved Ocular Oncology Biobank, (NRES REC Ref 16/NW/0380) sponsored by the University of Liverpool and with local approval in place at the Liverpool University Hospitals NHS Foundation Trust. Seventy-eight patients from TCGA data were analyzed for miR expression: 52 were non-metastatic and 26 were metastatic. TCGA-UM miR expression dataset was obtained from UCSC-Xenabrowser (https://xenabrowser.net/) deposited on The Cancer Genome Atlas data portal (TCGA). Level 3 data were downloaded from TCGA DCC. For each sample, all isoforms, expression of the same miRNA mature strand, are added together, log2 (total_RPM +1) transformed and deposited at UCSC into Xena repository, as described on https://xenabrowser.net. Two of the original 80 patients were excluded from the analysis because they were double mutated for GNAQ/GNA11. TCGA UM miR expression dataset were unlogged for further analysis.

### 4.3. RT-qPCR Analyses

Total RNA was extracted from cell lines using a miRNeasy mini kit (Qiagen, Hilden, Germany). Relative miR expression analysis was performed by RT-qPCR into Roche LightCycler 480 II using miScript SYBR Green Assay Kit and miScript primer assays (Qiagen).

The miScript primer assay in combination with miScript SYBR Green PCR kit from Qiagen was used for detection of miR122, 144, 221 and 222. RT-qPCR analysis for *ADAM10* and *c-Met* was performed as described [14]. Sequences of *ADAM10* primers were: GAGGAGTGTACGTGTGCCAGTT, Forward, GACCACTGAAGTGCCTACTCCA, Reverse; *c-Met* primers were: TGCACAGTTGGTCCTGCCATGA, Forward, CAGCCATAGGACCGTATTTCGG, Reverse. Relative quantification of mRNAs was calculated by the Ct method. Standardization of miRs between samples was obtained by using U6 small nuclear RNA as an internal control. Sequences of internal control primers were: TGACTTTGTCACAGCCCAAG Forward, TTCAAACCTCCATGATGCTG Reverse, for *B2M*, TGCCCTGAGGCACTCTTC Forward, TGAAGGTAGTTTCGTGGATGC Reverse for *ACTB*, and GACAATGCAGAGAAGCTGG Forward, GCAGGAAGACATCATCATCC Reverse for *RPII*. Standardization of *ADAM10* and *c-Met* genes was obtained by using -actin, 2-microglobulin and RNA polymerase II as internal control. Values are the mean ± sem of *n* = 3 independent RT-qPCR analyses.

### 4.4. Microarray and miRs Expression Analyses

miRCURY LNA probes (miRBase 9.2.) were spotted onto UltraGAPS II slides (Corning) by using the OmniGrid Accent spotter (GeneMachines, San Carlos, CA, USA) and crosslinked by UV [54]. miRCURY LNA microRNA Array Power labeling kit (Exiqon, Qiagen) was used for miRs labeling according to the recommendations of the manufacturer. For image acquisition, we used the Axon Genepix 4000B scanner (Molecular Devices, Sunnyvale, CA, USA) and dedicated software. Intensity data were preprocessed and normalized, to create a miR expression profile, following standard procedures as previously described [53]. MiR expression profiles were unlogged for further analysis.

### 4.5. MTT Assay

Cell proliferation was tested using a 3-(4,5-dimethylthiazol-2-yl)-2, 5-diphenyl tetrazolium bromide (MTT) assay, (Sigma-Aldrich, Milan, Italy). Optical density (OD) was read on a multiwell scanning spectrophotometer (ELISA reader BioTek, Savatec, Torino, Italy) at 570 nm.

### 4.6. MiRs Transfection

MiR mimics were obtained from Qiagen: miR122-5p (cat. no MSY0000421), miR144-3p (cat. no. MSY0000436). 92.1, UPMM3 and OMM2.5 cell lines were transfected with 100 nM mimic with lipofectamine 2000 (Invitrogen/ThermoFisher Scientific, Milan, Italy). AllStars (Qiagen) were used as negative control. After 48 or 72 h, cells were harvested and used for RNA extraction (mRNeasy Mini Kit, Qiagen) and real-time PCR analysis of miRs (miScript PCR System, Qiagen) or for Western blot analysis. Block IT Fluorescent Oligo (Invitrogen,) was also used to test efficiency of transfection by FACS analysis (FACScan BD, Becton & Dickinson Italy, Milan, Italy).

### 4.7. Western Blot

The Western blot of cell lysates was performed as previously described [14]. C-Met was detected by mouse anti-c-Met monoclonal antibody (Cell Signaling, Danvers, MA, USA), while ADAM10 was detected by rabbit anti-ADAM10 antibody (Abcam, Cambridge, UK). Anti-α-tubulin monoclonal and anti-GAPDH antibodies (Sigma-Aldrich, Milan, Italy) were also used as loading control.

### 4.8. Migration Assay

BD BioCoat migration chambers (BD Biosciences Italy, Milan, Italy) were used for migration assays, as described [55]. Cells were cultured in medium containing 1% FCS for 18 h before the assay. Recombinant HGF (100 ng/mL; Peprotech Tebu-Bio Italy, Milan, Italy) was added to the bottom chambers. After 24 h of incubation, migrated cells were fixed and stained with 1% toluidine blue. Migrating cells were calculated in a minimum of 10 fields, and the mean for each chamber determined. Experiments were run in triplicate.

### 4.9. Detection of Soluble c-Met

Soluble c-Met (sc-Met) was measured in the conditioned medium of cell lines upon cell debris removal by centrifugation at 3000 g for 20 min followed by a second centrifugation at 10,000 g for 30 min.

The level of sc-Met was measured using the Human Total HGFR/c-MET DuoSet IC enzyme-linked immunosorbent assay (ELISA) kit (R&D Systems, Abingdon, UK) following the manufacturer’s instructions. Optical density was measured at dual wavelengths of 450 and 540nm (400ATC; SLTLaboratory Instruments).

### 4.10. Annexin V-FITC/PI Staining Assay

1 × 10^5^ cells were centrifuged and resuspended in 500 μL of 1X binding buffer. Then Annexin V-fluorescein isothiocyanate (FITC) and Propidium Iodide (PI) were added accordingly with the manufacturer’s instructions (eBioscience, ThermoFisher Scientific, Milan, Italy). Flow cytometer (FACSCalibur; Becton Dickinson, East Rutherford, NJ, USA) was utilized to analyze cell apoptosis and CellQuest (Becton Dickinson) software was used for data analysis.

### 4.11. Cell Cycle Analysis

Transfected cells were washed twice in PBS, and permeabilized in cold 70% ethanol and incubated overnight at 4 °C in the dark. After washing twice with PBS, the cells were incubated in 1ml of PI staining solution for 30 min at room temperature in the dark. Biological role of irrelevant miRNA, miR122 and miR144 mimic in UM cells was determined using a flow cytometer (FACSCalibur; Becton Dickinson). The cell cycle was analyzed by ModFit (Becton Dickinson).

### 4.12. Luciferase Assay

92.1 cells were seeded in 96-well plates (2 × 10^4^ cells/well) for 24 h. For the reporter gene assay, cells were co-transfected with pGL3-c-Met-3’UTR (0.5 μg) or pGL3-ADAM10-3’UTR or empty control plasmid, and miR144 or miR122 mimic or irrelevant miR, in the presence of Renilla plasmid, by Lipofectamine 2000 (Invitrogen). The firefly and Renilla luciferase activities were measured using the dual luciferase assay (Promega, Milan, Italy) 48 h after transfection, and expressed as RLU (Relative Light Units). Experiments were run in triplicate and results were expressed as mean ± sem of three independent experiments.

### 4.13. Statistical Analyses

The paired or unpaired Student’s *t*-test was used when appropriate. P values are shown as following: * *p* < 0.05; ** *p* < 0.01; *** *p* < 0.001. Analyses were performed using PRISM 5 (Graph-Pad Software, San Diego, CA, USA).

### 4.14. Ethical Approval and Informed Consent

All experimental protocols were approved by institutional and licensing committees as detailed in the Methods Section 4.2. The methods were carried out in accordance with the relevant guidelines and institutional regulations, in accordance with the Declaration of Helsinki. Informed consent was obtained from all participants.

## 5. Conclusions

In this report, we show for the first time that miR122 and miR144 target directly the 3’UTR of the pro-invasive molecules *ADAM10* and *c-Met* in UM.

UM patients and cell lines display low levels of these miRs while expressing elevated levels of *ADAM10* and *c-Met*. These data suggest an oncosuppressive role of these miRs supported by the in vitro anti-proliferative and anti-migratory activity induced by their overexpression in UM cells. Further studies are needed to identify mechanisms that modulate the expression of miR122 and miR144 in UM. Xenograft models of UM will be necessary to study the effects of restoration of these suppressive miR122 and miR144 on UM growth and metastases formation. In vivo studies would provide the rationale for future therapeutic delivery of tumor suppressive miRs in UM patients.

## Figures and Tables

**Figure 1 cancers-12-01468-f001:**
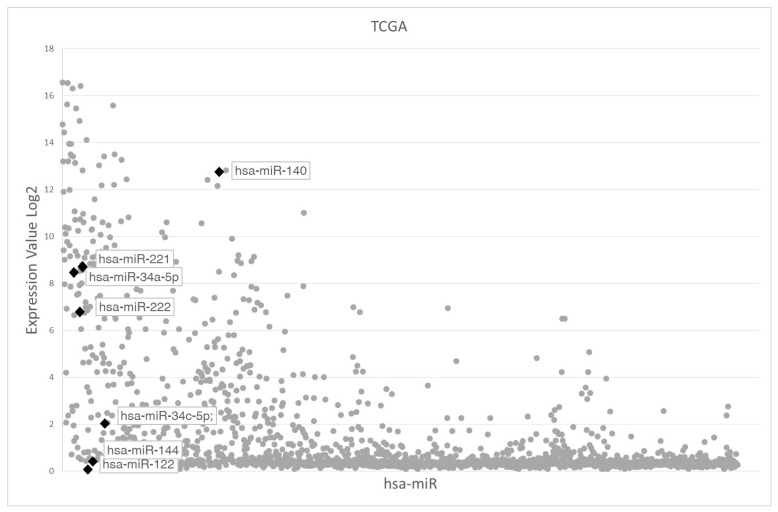
Scatter plot of miR expression in the uveal melanoma (UM) The Cancer Genome Atlas data portal (TCGA) dataset. *x*-axis: miRs (1938) present in the TCGA; *y*-axis: mean expression value of each miR in the 78 patients. MiR122 and miR144 are present at the bottom level of expression within all miRs. The expression of other miRs which target either *ADAM10* or *c-Met* is indicated for comparison. The expression levels of miR221 and miR222 are also indicated as positive control of miRs expression. The expression value is calculated as the base 2 logarithm of fluorescence intensity of each miR.

**Figure 2 cancers-12-01468-f002:**
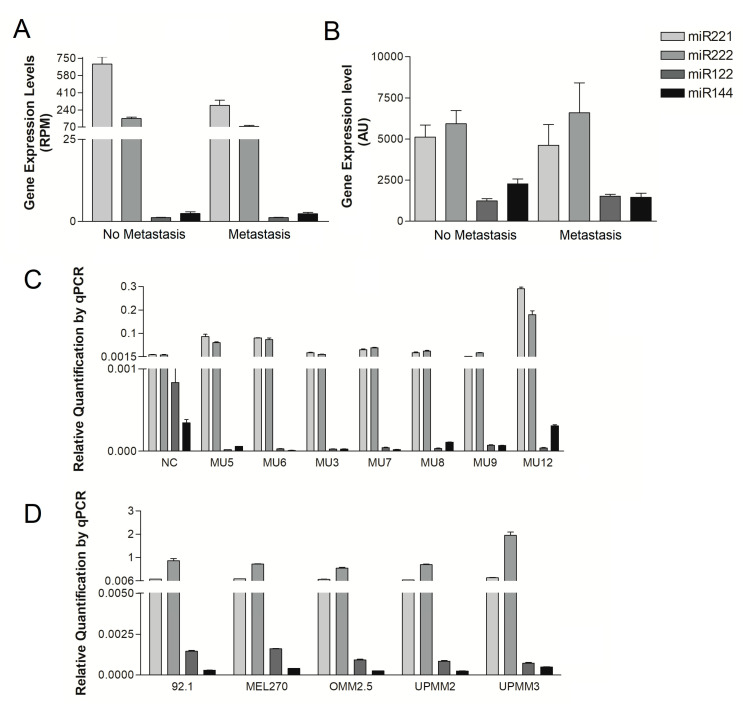
miR expression in UM patients and cell lines. (**A**): miR122, miR144, miR221 and miR222 expression in 78 UM from the TCGA: 52 UM were non-metastatic and 26 were metastatic. MiR levels are expressed as log2 (total_RPM +1) transformed. RPM: reads per million reads. (**B**): miR122, miR144, miR221 and miR222 expression in 19 primary UM, of which 5 were metastatic and 14 non-metastatic. Microarray data show miRNA normalized and log2 transformed of the fluorescence intensity (AU) (miR122 *p* = 0.66 and miR144 *p* = 0.77). (**C**): miR122, miR144, miR221 and miR222 expression in normal choroid (NC) and 7 UM using RT-qPCR. Only UM 5 and 6 were non-metastatic. Expression of miR122 and miR144 was lower in UM samples than in normal choroid. Data are presented as relative quantification analysis (1/delta Ct) with efficiency-method, compared to the U6 internal control. Values are the mean ± sd of *n* = 3 independent RT-qPCR analyses. (**D**): expression of miR122 and miR144 was measured in a panel of 5 UM cell lines using RT-qPCR. Data are presented as relative quantification analysis with efficiency-method, compared to the U6 internal control. Values are the mean ± sd of *n* = 3 independent RT-qPCR analyses. Light gray bars = miR221, gray bars = miR222, dark gray bars = miR122 and black bars = miR144.

**Figure 3 cancers-12-01468-f003:**
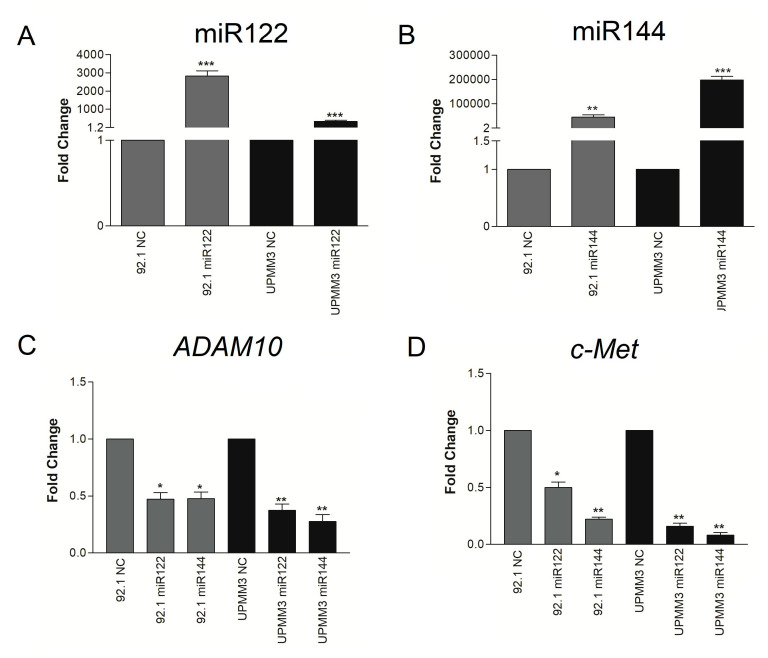
Effects of miR122 and miR144 mimic transfection in UM cell lines. MiR122 (**A**) and miR144 (**B**) expression increases in cell lines 92.1 and UPMM3 when transfected with the corresponding miR mimic. Increments in miR expression ranged from 800- to 197,000-fold, relative to transfection control (irrelevant miR: NC). MiR122 and miR144 mimic transfection inhibit *ADAM10* (**C**) and *c-Met* (**D**) mRNA expression levels, measured by RT-qPCR, in 92.1 and UPMM3 UM cell lines. Results are presented as fold change versus irrelevant miR control (NC) of relative quantification analysis efficiency-method based on miR122 and miR144 targets compared to the U6 internal control. Values are the mean ± sd of RT-qPCR analyses. Consistent data were obtained in two additional experiments (* *p* < 0.05; ** *p* < 0.01; *** *p* < 0.001).

**Figure 4 cancers-12-01468-f004:**
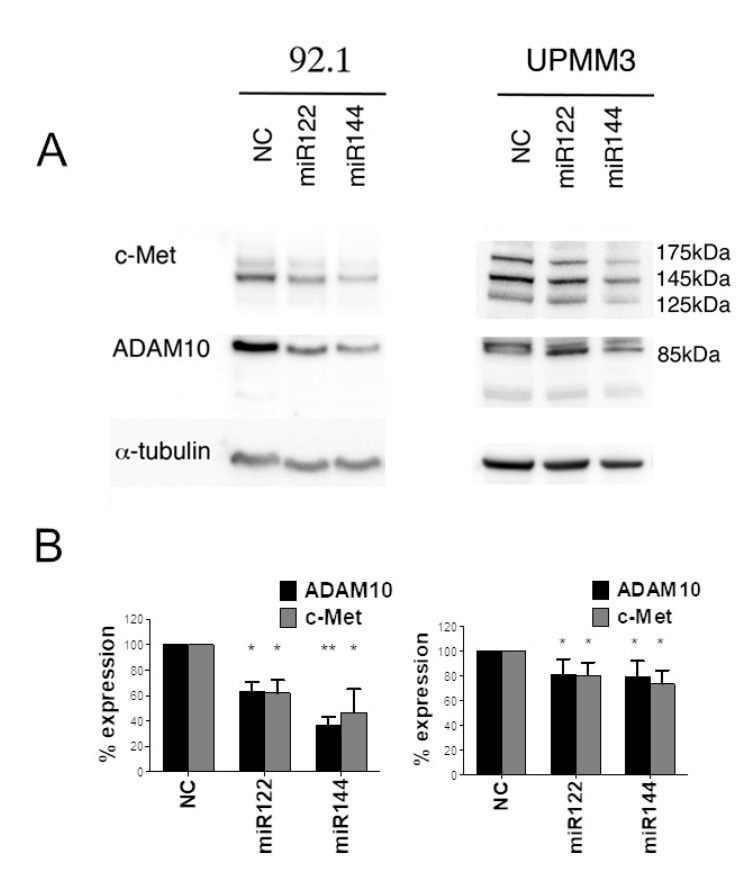
Western blot analysis of ADAM10 and c-Met expression of miR122 and miR144 transfected cells. (**A**): Western blot analysis shows expression of ADAM10 and c-Met proteins in 92.1 and UPMM3 cell lines, upon transfection with mimic miR122, miR144 or an irrelevant miR control (NC). α-tubulin was used as loading control. Cropped images are obtained from the same blot subsequently probed with the different antibodies. Molecular weight markers are shown. (**B**): histograms show the quantification of band intensity expressed as percentage of the irrelevant miR transfected control (NC). Mean ± sd of three independent experiments. (* *p* < 0.05; ** *p* < 0.01).

**Figure 5 cancers-12-01468-f005:**
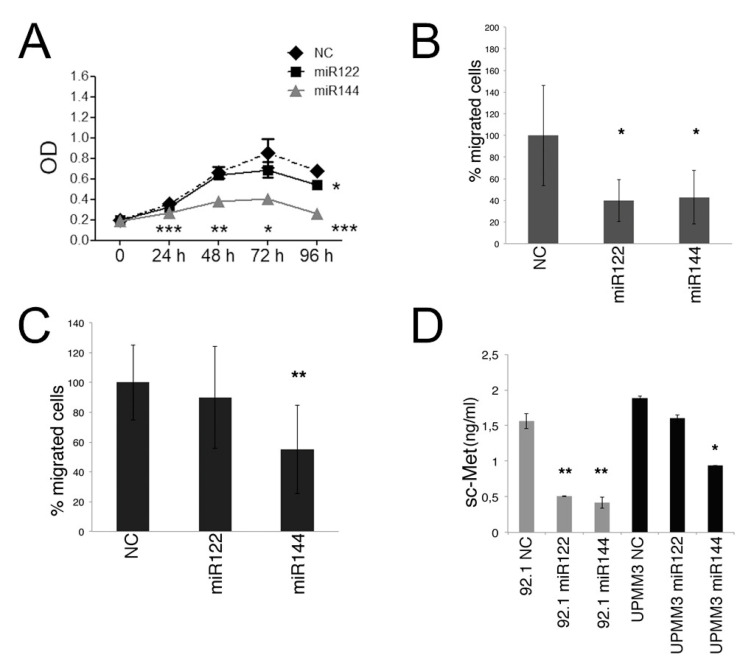
Effects of miR122 and miR144 mimic transfection on cell proliferation, migration and release of soluble c-Met. (**A**): proliferation, detected by a 3-(4,5-dimethylthiazol-2-yl)-2, 5-diphenyl tetrazolium bromide (MTT) assay, is significantly reduced in 92.1 cells transfected with miR122 and miR144 mimic. Data are expressed as mean ± sd of three different experiments. (**B**): migration assay shows a reduction of migrated 92.1 UM cells upon transfection with mimic miR122 or miR144, while migration of UPMM3 cells (**C**) is inhibited with mimic miR144. A representative experiment of three is shown. (**D**): transfection of mimic miR144 reduces soluble c-Met (sc-Met) levels in conditioned medium of the 92.1 and UPMM3 UM cell lines while miR122 affects soluble c-Met in 92.1 cells. Data are representative of three experiments (* *p* < 0.05; ** *p* < 0.01; *** *p* < 0.001).

**Figure 6 cancers-12-01468-f006:**
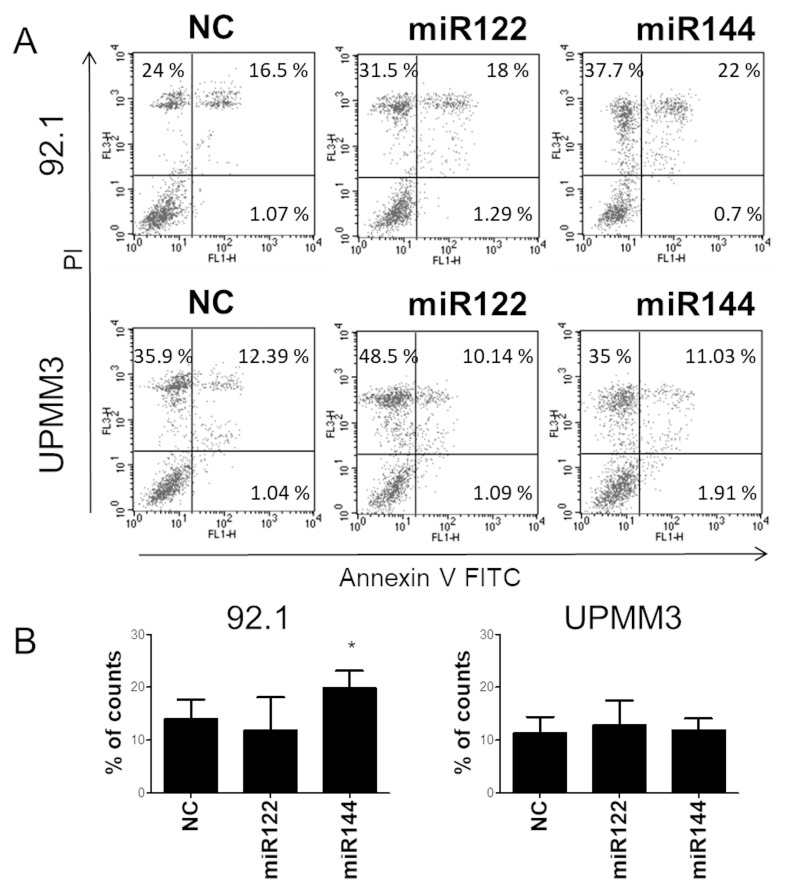
Flow-cytometric analysis of Annexin-V-PI staining of 92.1 and UPMM3 cells 72h after transfection with irrelevant miR (NC), miR122 or miR144 mimic. (**A**): a representative experiment is shown as dot plot; *x*-axis: Annexin-V-fluorescein isothiocyanate (FITC) staining, *y*-axis: Propidium Iodide (PI) staining. Viable cells take up no stains (lower left quadrant). Cells stained with PI alone are necrotic (upper left quadrant), cells stained only with Annexin-V-FITC constitute early apoptosis (lower right quadrant). Cells at final stages of apoptosis display both stains (upper right quadrant). Data are expressed as percentage of cells in each category. (**B**): Percentage of early apoptotic and apoptotic cells from three different experiments were used to calculate the mean ± sd of 92.1 and UPMM3 apoptotic cells (* *p* < 0.05).

**Figure 7 cancers-12-01468-f007:**
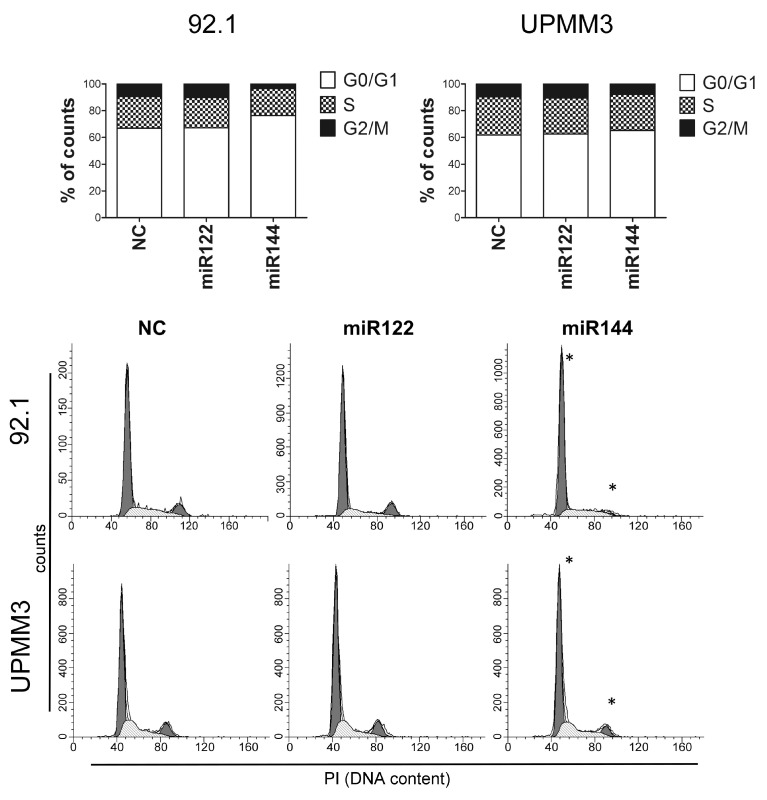
Cell cycle analysis of UM cells transfected with miRs mimic. Stacked plots are mean of three different experiments of 92.1 and UPMM3 cells transfected with irrelevant miR (NC), miR122 or miR144 mimic, 72 h after transfection. Standard deviation was below 5% of the mean. Representative cell cycle histogram analysis of 92.1 and UPMM3 miRs transfected cells shows a statistically significant increase in G0/G1 and decrease in G2/M phases in miR144 transfected cells (* *p* = 0.03). Histograms: *x*-axis: PI in FL2-Area (DNA content), *y*-axis: counts.

**Figure 8 cancers-12-01468-f008:**
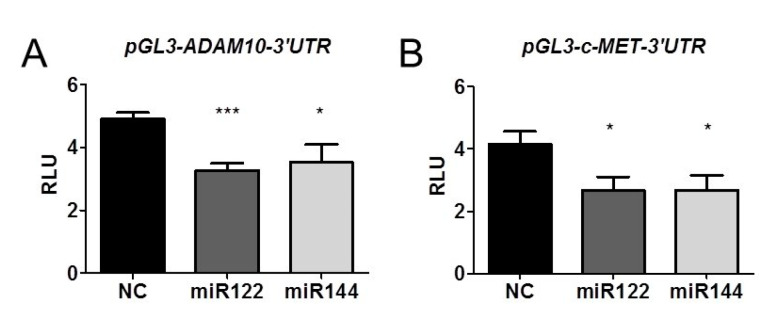
MiR122 and miR144 directly target the *ADAM10* and *c-Met* 3’UTR. Co-transfection of miR122 or miR144 mimic with either pGL3-ADAM10-3’UTR (**A**) or pGL3-c-Met-3’UTR (**B**) reduces luciferase activity in 92.1 cells. Irrelevant control miR (NC) does not affect luciferase activity of pGL3-c-Met-3’UTR or pGL3-ADAM10-3’UTR. RLU (Relative Light Units) are obtained normalizing Luciferase activity to Renilla. Data represent mean ± sem. Data are representative of three experiments (* *p* < 0.05; *** *p* < 0.001).

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
