# Peer review of "Potential Onco-Suppressive Role of miR122 and miR144 in Uveal Melanoma through ADAM10 and C-Met Inhibition"

_cancers, 2020, doi:10.3390/cancers12061468_

Round 1
Reviewer 1 Report
Authors provide intesting in vitro work and have addressed concerns raised to the lack of evidence of the presented microRNAs to ADAM10 / c-MET expression sufficiently. I have no further comments
Author Response
REV 1
Comments and Suggestions for Authors
Authors provide intesting in vitro work and have addressed concerns raised to the lack of evidence of the presented microRNAs to ADAM10 / c-MET expression sufficiently. I have no further comments
Response: We thank the Reviewer for his/her time; his/her comments have helped to improved our manuscript.
Reviewer 2 Report
This reviewed manuscript is acceptable at the present status.Author Response
REV 2
Comments and Suggestions for Authors
This reviewed manuscript is acceptable at the present status.
Response: We thank the Reviewer for his/her time; his/her comments have helped to improve our manuscript.
Reviewer 3 Report
The Authors have addressed my comment in a satisfactory way. However, I have additional concerns about statistics.
- Fig. 2 would benefit if the Authors perform statistical analysis. It is necessary to support statement e.g., in lines 114-115.
- It is still confusing to state *P =0.05 (e.g., Fig. 3, 5). Are the Authors sure that p value was EQUAL 0.05? I suppose there should be p<0.05.
- Fig. 4 "P =/<0.05" - what does it mean?
- In addition, Annexin-V/PI staining in the Fig. 6 is doubtful. Firstly, axis should be named (PI, Annexin-V). Secondly, there are substantial percentages of PI-positive in controls. Thirdly, "apoptotic cells" - how were they defined? Are they Annexin-V-positive cells? Fourthly, dot plots, not histogram plots, are shown.
- Fig. 7 is of poor quality.
Author Response
REV 3
Comments and Suggestions for Authors
The Authors have addressed my comment in a satisfactory way. However, I have additional concerns about statistics.
Response: We thank the Reviewer for his/her time and comments. We followed all suggestions and hope that the Reviewer agrees with the changes. Please find the revisions in the manuscript highlighted in green and below as responses to the Reviewer
1. Fig. 2 would benefit if the Authors perform statistical analysis. It is necessary to support statement e.g., in lines 114-115.
Response: An unpaired Student’s t-test had been performed as described in Methods, lines 437-439. The results showed lack of significant differences (NS). We now added the P values, both in the text (line: 120 (miR122 P=0.66 and miR144 P=0.77) ) and in the legend of Figure 2 (line: 151 (miR122 P=0.66 and miR144 P=0.77) )
2. It is still confusing to state *P =0.05 (e.g., Fig. 3, 5). Are the Authors sure that p value was EQUAL 0.05? I suppose there should be p<0.05.
Response: The reviewer is right, we corrected *P =0.05 in *P <0.05 legend to Figure 3 (line 182) and 5 (line 218-219)
3. Fig. 4 "P =/<0.05" - what does it mean?
Response: we corrected *P <0.05 in the legend to Figure 4 (line 196)
4. In addition, Annexin-V/PI staining in the Fig. 6 is doubtful. Firstly, axis should be named (PI, Annexin-V). Secondly, there are substantial percentages of PI-positive in controls. Thirdly, "apoptotic cells" - how were they defined? Are they Annexin-V-positive cells? Fourthly, dot plots, not histogram plots, are shown.
Response: we modified Figure 6 as suggested:
1.The Referee is right. Now the axes are defined: x-axis: Annexin V FITC; y-axis: PI, both in Figure 6 and in the legend to Figure 6 (line: 234)
2. Frequently adherent cell lines exhibit some positive PI staining, possibly related to membrane damage during the harvesting process. This quote of PI positive (necrotic, upper left quadrant) was reproducibly present in these cell lines. That is the reason why we have chosen to consider only Annexin V positive cells as the real apoptotic cells.
3.Apoptotic cells are all Annexin V positive (double-positive Annexin V/PI cells+ single positive Annexin V cells). We have now added this information to the legend to Figure 6 (line: 235-240)
4.We agree with the Referee that Figure 6 was not clear. Therefore we changed it. Now panel A shows a representative experiment as dot plot. The x-axis represents Annexin-V FITC staining. The y-axis represents PI staining. Viable cells take up no stains (lower left quadrant). Cells stained with PI alone are necrotic (upper left quadrant), cells stained with Annexin V-FITC alone represent early apoptosis (lower right quadrant). Cells at final stages of apoptosis take up both stains (upper right quadrant). Data are expressed as percentage of cells in each category. Panel B: Percentage of early apoptotic and apoptotic cells from three different experiments were used to calculate the mean ± sd of 92.1 and UPMM3 apoptotic cells, (*P<0.05) and are presented as histogram. This is now described in the Legend to Figure 6.
5. Fig. 7 is of poor quality.
Response: we edited Figure 7. In the legend to Figure 7, we added the following: Histograms: x-axis: PI (DNA content); y-axis: counts (line 251).
Round 2
Reviewer 3 Report
None remaining
This manuscript is a resubmission of an earlier submission. The following is a list of the peer review reports and author responses from that submission.
Round 1
Reviewer 1 Report
Authors hypothesize the potential onco-suppressive role of two micro RNAs: miR122 and miR144. These two miRNAs are involved in the post translational modification and could contribute to the down regulation of two genes ADAM10 and c-MET. Previously, this group found these two proteins to be expressed in 60% of UM and that silencing of ADAM10 reduces the levels of soluble c-Met and inhibit cell invasiveness. ADAM10 and c-MET gene expression correlated with poor progression‐free survival. In this study, they investigate and show an effect of miR122 and miR144 with in vitro studies in a UM cell line.
The manuscript text is well-written and the in-vitro work is interesting. My major concern is the lack of evidence presented linking these two microRNAs to ADAM10 / c-MET expression in patients and to their effect in patients at risk for metastasis.
Major comments
Authors show a low expression of these two miRNAs and analysis of both TCGA and their own dataset do not indicate a difference between patients with or without metastasis. [results line 113-114] Is there a difference in c-MET/ADAM10 expression in patients with higher miR122/144 compared to patients with lower miR122/144? I am missing the evidence of the effect of this difference in UM patient data/tumours.
In their in-vitro experiments (in UM cell lines) authors find a difference in mRNA levels between cell lines 92.1 (no chr 3 loss, BAP1 wt, EIFAX mutated) and UPMM3 (chr 3 loss, BAP1 mutated). The reduction of high ADAM10 / c-MET by miR122 /144 would indicate a possible onco-suppressive effect. However, on protein level this difference is only seen in 92.1 and not UPMM3 (which would be prone to metastasis). This is a bit odd as one would hope for an onco-suppressive effect in a tumour that would metastasize (chr3 loss/bap1 mutated) on perhaps not in one that rarely metastasizes. A similar effect is seen on soluble c-MET (figure 5a) and cell migration (figure 5c/d). The effects is larger in the 92.1 line compared to the UPMM3 line.
Why is the effect of these two miRNAs not clearly seen in the in vitro model for a metastasizing tumour? What does this mean for the importance of these two micro RNAs as a onco-suppressor?
Minor comments:
Line 71-72 “strongly correlated” to what?
Figure 2: what do the different shades of grey represent? Are these the in the same order as mentioned in the text? What is statistical different? I assume the control micro RNAs and Mir122 and miR144 but this is not clear from the picture.
Figure 2 A/B/D Furthermore, authors have chosen the controls based on this difference so it is not clear to me why they indicate this in the picture. This picture represented the absence of difference between patients with and without a metastasis (A/B) or expression levels in cell lines (D)
Expression levels are indicated, there is no mention of how it is represented (RKPM, TPM??) Additionally, they levels in the TCGA set differ about a 100 fold with those in the dataset used in B. Please elaborate why.
Figure 2C/D. The 1/delta ct levels are not clearly depicted. (somewhere between 0.005 , 0.001 and 1)
Figure 4 Author state that these are cropped images of the same western blot (line 176-177). Why do the number of band differ between 92.1 and UPMM3?
In general : figures are difficult to interpret as not all information necessary to interpret the figures are present in the accompanying figure text. (it is present in the manuscript text).
Reviewer 2 Report
The authors did not demonstrate enough evidence to support their hypothesis. At least, they need to do more cell function analysis including the experiment in vivo.
Therefore, I think this manuscript is not acceptable.
Reviewer 3 Report
The study by Amaro et al. demonstrates that miR-122 and miR-144 display a putative onco-suppressive role in uveal melanoma through ADAM10 and c-Met modulation. The results could be presented in a more encouraging way. Although the study provides some novel observations, in general presented data have been already reported by others.
Following concerns need to be addressed.
1) Certain statements need to be re-written or require English corrections, eg. line 50 (abstract), line 102.
2) Figure 2 is impossible to be understood. What do the colors mean? Any of four miRs? If so, there must be a legend! In addition, using "Met" abbreviation for "metastasis" in case of studying the role of c-MET receptor is not a good idea. Finally, Fig. 2A and B - how the gene expression was calculated/expressed?
3) The Authors mention different genetic alterations typical of specific cell lines eg. chromosome-3 monosomy or BAP1 mutational status. It would be worth to discuss putative role of these alterations with regard to the subject of this study. Especially when you consider the differences eg., in the Fig. 4.
4) I suppose this is not true: "p*=0.05; p**=0.002; p***=0.0001"
5) Fig. 4 please provide quantification as average measurement of three replicates.
6) For Fig. 5B, OD at the zero timepoint should be included. It is not obvious that each type of cells were seeded equally.
7) c-MET can also regulate cell survival. Did the Authors perform any assessment of apoptosis? MTT assay that was used in this study and is described as proliferation assay, provides actually data that are a collective effect of cell proliferation and cell death.
8) 3.3 - forward and reverse primers rather than lower and upper is recommended. The primer sequences for internal controls should be given as well.
